# Right Ventricular Endocardial Mapping and a Potential Arrhythmogenic Substrate in Cardiac Amyloidosis—Role of ICD

**DOI:** 10.3390/ijerph182111631

**Published:** 2021-11-05

**Authors:** Aleksandra Liżewska-Springer, Tomasz Królak, Karolina Dorniak, Maciej Kempa, Alicja Dąbrowska-Kugacka, Grzegorz Sławiński, Ewa Lewicka

**Affiliations:** 1Department of Cardiology and Electrotherapy, Medical University of Gdansk, 80-952 Gdansk, Poland; tomasz.krolak@gumed.edu.pl (T.K.); maciej.kempa@gumed.edu.pl (M.K.); alidab@gumed.edu.pl (A.D.-K.); lek.grzegorzslawinski@gmail.com (G.S.); elew@gumed.edu.pl (E.L.); 2Department of Nonivasive Cardiac Diagnostics, Medical University of Gdansk, 80-952 Gdansk, Poland; karolina.dorniak@gumed.edu.pl

**Keywords:** cardiac amyloidosis, implantable cardioverter-defibrillator, sudden cardiac death, electrophysiological study mapping

## Abstract

Patients with cardiac amyloidosis (CA) have an increased risk of sudden cardiac death. (SCD). However, the role of an implantable cardioverter-defibrillator in the primary prevention of SCD in this group of patients is still controversial. We present a case with CA with recurrent syncope and non-sustained ventricular tachycardia. In order to further stratify the risk of SCD, an electrophysiological study with endocardial electroanatomic voltage mapping was performed prior to the ICD placement.

## 1. Introduction

We present a patient with cardiac amyloidosis (CA) resulting from clonal production of immunoglobulin light chains (AL amyloidosis), with recurrent syncope and non-sustained ventricular tachycardia (nsVT), who underwent electrophysiological study with endocardial electro-anatomic voltage mapping, and was finally referred for implantable cardioverter-defibrillator (ICD) placement for primary prevention of sudden cardiac death (SCD).

Two main types of CA are: immunoglobulin light chain (AL) amyloidosis (AL-CA) and transthyretin (ATTR) CA. There has been a steady increase in the diagnosis of CA over the past decade. This is largely due to the development of non-invasive imaging modalities, such as strain echocardiography and scintigraphic nuclear imaging. The prevalence of AL-CA is 8–12 per million per year and it accounts for almost 70% of all newly diagnosed patients with CA [1].

The prognosis in CA is poor, patients typically show symptoms and signs of progressive heart failure. Cardiac arrhythmias, particularly atrial fibrillation are well-documented, but there is less data on ventricular arrhythmias. It has been reported that SCD accounts for up to 50% of all cardiac-related deaths in CA patients [2]. However, a study of AL-CA patients with implanted cardiac rhythm recorder revealed bradycardia with subsequent pulseless electrical activity (PEA) as a terminal rhythm in 62% of deaths [3]. Among 272 loop recordings, only one nsVT was found.

Therefore, the role of ICDs in the primary prevention of SCD in patients with CA is still controversial. On the one hand, there is an increased risk of SCD in CA patients, but other studies do not confirm that they benefit from ICD. It has been most frequently reported in the literature that, despite fairly frequent adequate interventions, ICD therapy does not provide a survival benefit in this patient group. Ventricular arrhythmias can be life-threatening, but these are the electromechanical disruptions or conduction disturbances that are considered the most common causes of SCD in patients with CA.

Thus, the role of ICD is uncertain and there are no clear guidelines for such treatment in CA patients. To date, several factors (including cardiac biomarkers and renal function parameters) have been documented to predict overall mortality. However, little is known about risk factors for the arrhythmic cause of SCD in CA patients. Furthermore, little is known about the pathophysiology of ventricular arrhythmias in CA patients, including invasive electrophysiological studies, which is essential for risk assessment [2].

In the presented patient, the endocardial voltage mapping was performed for the first time, which may be important for understanding the causes of arrhythmias in these patients.

## 2. History of Presentation

A 51-year-old Caucasian male was admitted to the cardiology department for syncope episodes and nsVT recorded on 7-day ambulatory Holter electrocardiographic (ECG) monitoring. On admission, the patient presented with symptoms of heart failure of class II according to the New York Heart Association (NYHA) classification, and occasional chest pain during strenuous exercise. He reported three episodes of syncope during normal activity in the last four months, the course of which could suggest arrhythmias or orthostatic hypotension. Pulse rate was 68 bpm with a regular rhythm, systemic blood pressure 108/69 mm·Hg, a third heart sound was audible on auscultation, but there were no signs of lung congestion or peripheral edema. The medications the patient was taking were diuretics (furosemidum 2 × 40 mg daily, spironolactone 50 mg daily) and ramipril 5 mg daily. The latter was withdrawn on admission due to low BP.

### 2.1. Past Medical History

Fifteen months earlier this patient was diagnosed with multiple myeloma and concomitant renal and CA resulting from AL fibrils deposit composed of monoclonal immunoglobulin lambda-type light chains (AL amyloidosis). The abdominal fat tissue biopsy revealed amyloid deposits after Congo red staining. Transthoracic echocardiography (TTE) showed biatrial enlargement, concentric left ventricular (LV) and right ventricular (RV) myocardial hypertrophy, preserved LV ejection fraction (LVEF) of 55%, and LV diastolic dysfunction. Cardiac magnetic resonance imaging (cMRI) revealed a spectrum of typical features of CA. The patient was referred for chemotherapy and underwent four cycles according to the MPV regimen (melphalan, prednisone, bortezomib), and later the treatment was changed to VCD (bortezomib, cyclophosphamide, dexamethasone). After one year of treatment, based on the haematological response criteria, a good partial response was obtained (free light chain normalization, significant reduction of proteinuria).

### 2.2. Investigations

Upon admission blood tests revealed an increased N-terminal pro-brain natriuretic peptide (NT-proBNP) level of 1700 pg/mL (normal: <40 pg/mL) and hs-cTnI concentration of 0.134 ng/mL (normal: <0.034 ng/mL), creatinine level was 1.08 and eGFR 79 mL/min/1.73 m^2^ and electrolyte levels were normal.^.^ The patient was classified in stage III according to the Mayo 2004 staging system [4]. ECG showed sinus rhythm with low voltage QRS complexes in limb leads, right axis deviation and right bundle branch block. There was no narrowing of coronary arteries in coronary computed tomography angiography, and the Agatston score was 0. Seven-day ambulatory Holter ECG monitoring revealed three nsVT episodes with two different morphologies (the fastest of 169 bpm, and the longest of 7 s and including 20 beats). During Holter monitoring, the patient reported dizziness and worsening of exercise intolerance. TTE demonstrated, as previously, enlargement of the left (36 mL/m^2^) and right atrium (28 cm^2^), LV hypertrophy (interventricular septum up to 14 mm), mildly reduced LVEF of 50%, a restrictive filling pattern and a significantly abnormal LV global longitudinal strain (GLS) of −10.4% with an apical sparing pattern (Figure 1A,B).

The cMRI showed very high native T_1_ and T_2_ relaxation times, markedly increased extracellular volume fraction (ECV) of 70% and generalized transmural late gadolinium enhancement (LGE) involving the entire LV and RV myocardium as well as the atrial walls (Figure 1C,D). Microvolt T-wave alternans (MTWA) testing was performed on a treadmill, and the result was negative. After approval by a local ethics committee and the patient’s written informed consent was obtained (ethical approval reference number: NKBBN/725/2020), an invasive electrophysiology study (EPS) was performed and revealed abnormal sinoatrial and atrioventricular (AV) conduction. The corrected sinus node recovery time (cSNRT) was significantly abnormal: with the first pause at 4.6 s and the second pause at 5.5 s (normal value < 1.5 s). The His bundle-ventricular (HV) interval was prolonged to 62 ms (normal range 35–55 ms). Programmed ventricular stimulation was performed in accordance with the local protocol: up to three extrastimuli at two paced cycle lengths: 600 ms and 400 ms and the pacing site was the RV apex. No arrhythmia was induced. Then, bipolar endocardial electro-anatomic voltage mapping of RV and right atrium was performed with the use of ThermoCool SmartTouch catheter (Biosense Webster Inc., Irvine, CA, USA) and 3D electroanatomical system CARTO 3. It revealed normal voltage of endocardial potentials and no low voltage areas were recorded (Figure 1E). This image was similar to that of a patient without any heart disease, who had intracardiac right heart mapping prior to ablation for ventricular arrhythmia originated from the RV outflow tract (Figure 1F).

Finally, a dual-chamber ICD was inserted (Rivacor 5 DR-T, Biotronik, Germany). During the 12-month follow-up, there were no episodes of arrhythmia in the routine ICD controls.

Currently, the patient is after an autologous stem-cell transplant.

## 3. Discussion

Among patients with CA, SCD accounts for up to 50% of all cardiac deaths [5]. However, the frequency of arrhythmic SCD is unknown, and electromechanical dissociation or (less commonly) AV conduction disturbances are considered the most common cause of SCD in CA patients. nsVT is commonly found in patients with CA, but has a low predictive value as a risk factor for malignant ventricular arrhythmias [2]. Consequently, the role of ICD is controversial in this group of patients and there are currently no European guidelines on ICD treatment for primary prevention in patients with CA. According to the 2019 Heart Rhythm Society consensus statement, a prophylactic ICD implantation may be considered in patients with AL cardiac amyloidosis and nsVT in whom the expected survival is longer than 1 year [6]. However, this is only a class IIb recommendation. Furthermore, a syncope is a common finding in CA patients, but is a non-specific symptom and may result from various causes, not only arrhythmias, such as orthostatic hypotension, autonomic dysfunction, the use of diuretics or vasodilating drugs, and AV conduction disturbances [2].

In the presented patient, EPS was performed in order to further stratify the risk of SCD. It should be emphasized that EPS is rarely performed in CA patients, and we found only two studies reporting electrophysiological abnormalities among CA patients in EPS. So far, there are no studies that have performed intracardiac mapping in this patient group. Reisinger at al. [5] indicated that markedly prolonged HV interval (≥80 ms) in patients with CA was the only independent predictor for SCD; however, this did not occur in the presented case (HV interval of 62 ms). The HV interval > 55 ms is frequently found in CA patients [2,5]. Its significant prolongation (≥80 ms) may indicate, on the one hand, a risk of complete AV block occurrence due to amyloid infiltration of the conduction system, and on the other hand, a significant infiltration of the myocardium by amyloid fibrils and thus an increased risk of death due to electromechanical dissociation or ventricular arrhythmias [5]. However, ventricular tachyarrhythmias are rarely induced in EPS in CA patients [2,5] and also have not been induced in the presented patient. In turn, low-voltage areas detected during endocardial electro-anatomical mapping may indicate the presence of potential arrhythmogenic substrate for ventricular tachycardia (VT) [7]. However, to our surprise, the results of right heart voltage mapping our patient did not reveal any abnormalities, suggesting that the substrate for VT (if any) may be difficult to identify. Additionally, we found no correlation between the electro-anatomical mapping (absence of low voltage areas in the right heart) and LGE (diffuse, involving most of the myocardium). Furthermore, the MTWA testing, which was previously considered a predictor of increased risk of SCD in patients with heart failure, was negative in the presented case [8]. Nevertheless, based on the current knowledge and the proposed algorithm (Figure 2), the final decision was to implant an ICD. The patient reported recurrent syncope, nsVT was documented, but was in a relatively early stage of heart disease as indicated by the mildly elevated cardiac biomarkers and preserved LVEF. However, the patient was diagnosed with reduced LV GLS and diffuse transmural LGE, which are considered markers potentially identifying CA patients that may benefit from ICD implantation [2].

On the other hand, there were no abnormalities in endocardial voltage mapping of RV and right atrium, and supposing that in AL amyloidosis, amyloid fibrils infiltrate both left and right ventricles, it can be assumed that the result of LV endocardial mapping would be similar. The significance of these findings is evidenced by the fact that during the 12-month follow-up, the patient did not develop any ventricular arrhythmias. However, the occurrence of arrhythmias with a slower rate than the programmed detection rate of the ICD cannot be excluded. This could potentially cause an underestimation of arrhythmia burden. Therefore, EPS testing may be considered to better stratify the risk of SCD in patients with CA and nsVT.

## 4. Conclusions

Data on the role of EPS in CA patients are limited. Normal voltage of endocardial potentials, found for the first time in a patient with AL amyloidosis, in combination with a negative result of programmed ventricular stimulation, may indicate a small potential arrhythmogenic substrate in patients with CA. This is in line with the observation that patients with CA do not benefit from ICD for primary prevention. Further prospective studies are needed to understand the pathophysiology of arrhythmias in CA patients and thus to better stratify the risk of arrhythmic SCD.

## Figures and Tables

**Figure 1 ijerph-18-11631-f001:**
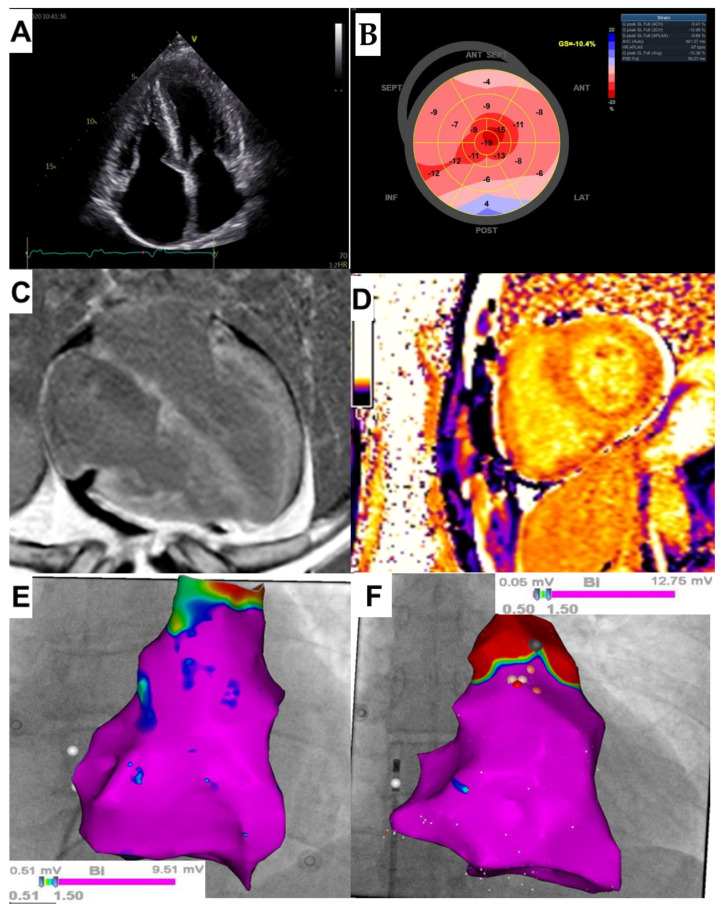
(**A**) Transthoracic echocardiography (TTE Vivid S95, GE Healthcare, Chicago, IL, USA): 4- chamber apical view demonstrating ventricular hypertrophy and atrial enlargement; (**B**) TTE (Vivid S95, GE Healthcare, Chicago, IL, USA): Left ventricular (LV) global longitudinal strain showing LV “apical sparring” pattern; (**C**) Cardiac magnetic resonance imaging (cMRI; Siemens Aera, 1.5 T, Erlangen, Germany): Phase-sensitive inversion recovery sequence showing the extent and distribution of late gadolinium enhancement (LGE) in the 4-chamber plane. Generalized, diffuse LGE is shown, involving the entire left and right ventricles as well as atrial walls; (**D**) cMRI (Siemens Aera, 1.5 T, Erlangen, Germany): Modified Look-Locker inversion recovery (MOLLI) sequence showing very low post-contrast T1 values, similar to the blood pool values, corresponding to the markedly increased myocardial extracellular volume (ECV) of 70% (reference range: 26 ± 3%); (**E**,**F**) Bipolar endocardial voltage map of the right ventricle (RV) (3D electroanatomical system CARTO 3, ThermoCool SmartTouch catheter -Biosense Webster Inc, Irvine, CA, USA) obtained in the presented patient with AL cardiac amyloidosis (**E**), and in the patient without any heart disease (**F**). Both patients exhibit normal myocardial bipolar voltages as indicated by a purple color representing normal myocardium with a voltage > 1.5 mV.

**Figure 2 ijerph-18-11631-f002:**
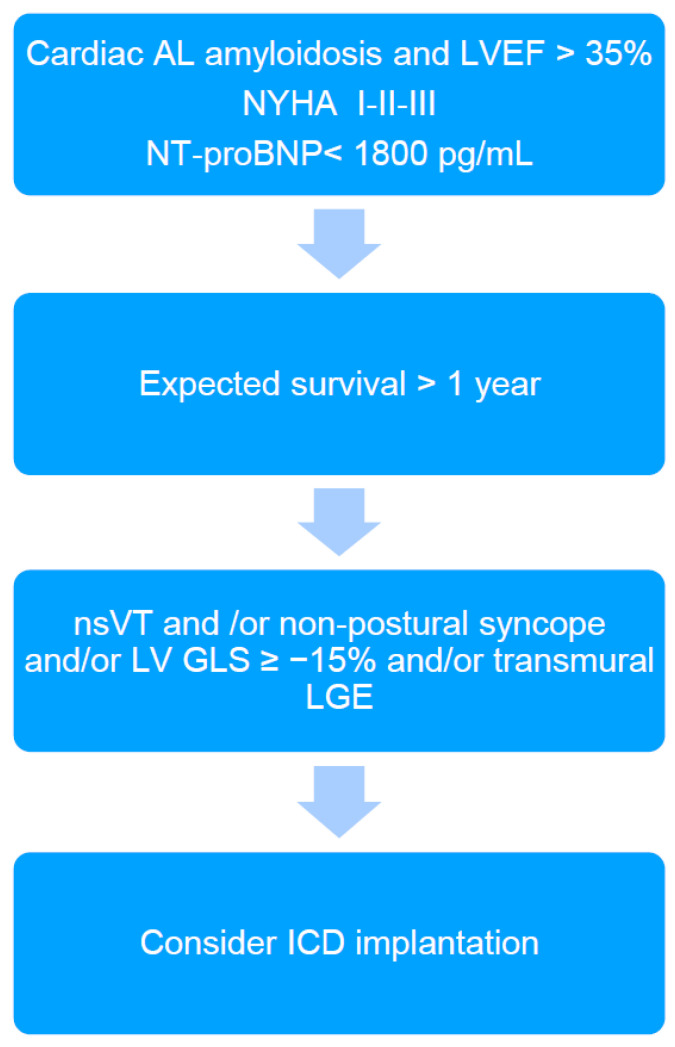
The algorithm proposed for qualifying cardiac immunoglobulin-derived light chains amyloidosis (AL) patients for ICD implantation in primary prevention of sudden cardiac death (2). ICD: implantable cardioverter-defibrillator; LVEF: left ventricular ejection fraction; NT-proBNP: N-terminal pro brain natriuretic peptide; nsVT: non-sustained ventricular tachycardia; LV GLS: left ventricular global longitudinal strain; LGE: late gadolinium enhancement.

## Data Availability

Not applicable.

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
