# Peer review of "Right Ventricular Endocardial Mapping and a Potential Arrhythmogenic Substrate in Cardiac Amyloidosis—Role of ICD"

_ijerph, 2021, doi:10.3390/ijerph182111631_

Round 1

Reviewer 1 Report

The role of ICDs in primary prevention of SCD in patients with CA is still controversial. On the one hand, there is an increased risk of SCD in CA patients, but other studies do not confirm that they benefit from ICD. In the presented patient, the endocardial voltage mapping was performed for the first time, which may be important for understanding the causes of arrhythmias in these patients.  

I believe and think that it could be valuable to insert by rephrasing as follows:  

The role of ICDs in primary prevention of SCD in patients with CA is still controversial. On the one hand, there is an increased risk of SCD in CA patients, but other studies do not confirm that they benefit from ICD .... [here briefly ... mention why the controversy, for example, how does the SCD (signs and symptoms, mentioning associated risk factors, among others), understanding that arrhythmias should be mentioned because at the end of this paragraph emphasis is placed on them, in the same sense, mention in which cases the ICD is used, also taking up everything As mentioned in more detail in the discussion, this may help to better understand this important clinical approach] ... In the presented patient, the endocardial voltage mapping was performed for the first time, which may be important for understanding the causes of arrhythmias in these patients.  

Even at the beginning of the Introduction, it should be mentioned in general how common the presence of cardiac amyloidosis is, this would give a justification that could round off the importance of reporting these clinical cases.    

Thank you for your attention, I remain at your service for future collaborations, greetings.

Reviewer 2 Report

The authors presented a case of AL amyloidosis, recurrent syncope and nsVT who underwent implantation of a dual chamber ICD. 

My comments are the following:

  1. How often is it a patient with AL amyloidosis to present with sinus rhythm?
  2. If the electroanatomical mapping and programmed ventricular stimuli were normal why was it decided the implantation of an ICD?
  3. Could the episodes of nsVT in Holter ECG be due to electrolytic disturbances?
  4. Were the recorded arrythmic episodes be followed by syncope?
  5. How the authors explain the absence of any recorded arrythmia at the follow up?
  6. It should be made clear by the authors if it is suggested a EPS study for those patients.

Round 2

Reviewer 2 Report

All my comments have been addressed. The manuscripts has been substantially improved